# Adaptive tip-enhanced nano-spectroscopy

Dong Yun Lee[1], Chulho Park[2], Jinseong Choi[1], Yeonjeong Koo[1], Mingu Kang[1], Mun Seok Jeong[3,4], Markus B. Raschke[5,6,7 ✉] & Kyoung-Duck Park [1✉]

Tip-enhanced nano-spectroscopy, such as tip-enhanced photoluminescence (TEPL) and tip-enhanced Raman spectroscopy (TERS), generally suffers from inconsistent signal enhancement and difficulty in polarization-resolved measurement. To address this problem, we present adaptive tip-enhanced nano-spectroscopy optimizing the nano-optical vector-field at the tip apex. Specifically, we demonstrate dynamic wavefront shaping of the excitation field to effectively couple light to the tip and adaptively control for enhanced sensitivity and polarization-controlled TEPL and TERS. Employing a sequence feedback algorithm, we achieve ~$4.4 \times 10^4$-fold TEPL enhancement of a $WSe_2$ monolayer which is >2× larger than the normal TEPL intensity without wavefront shaping. In addition, with dynamical near-field polarization control in TERS, we demonstrate the investigation of conformational heterogeneity of brilliant cresyl blue molecules and the controllable observation of IR-active modes due to a large gradient field effect. Adaptive tip-enhanced nano-spectroscopy thus provides for a systematic approach towards computational nanoscopy making optical nano-imaging more robust and widely deployable.

[1] Department of Physics, Ulsan National Institute of Science and Technology (UNIST), Ulsan, Republic of Korea. [2] Department of Energy Science, Sungkyunkwan University (SKKU), Suwon, Republic of Korea. [3] Department of Physics, Hanyang University, Seoul, Republic of Korea. [4] Department of Energy Engineering, Hanyang University, Seoul, Republic of Korea. [5] Department of Physics, University of Colorado, Boulder, CO, USA. [6] Department of Chemistry, University of Colorado, Boulder, CO, USA. [7] JILA, University of Colorado, Boulder, CO, USA. ✉email: markus.raschke@colorado.edu; kdpark@unist.ac.kr

Despite a range of groundbreaking discoveries with advances in tip-enhanced Raman spectroscopy (TERS) and tip-enhanced photoluminescence (TEPL), even in the strong coupling regime[1,2] and with atomic-resolution[3–5], many practical challenges still remain for facile and widespread implementation of tip-enhanced nano-spectroscopy. Even under the same excitation conditions, TERS or TEPL enhancement factor is non-uniform for different plasmonic tips since their apex size, shape, and surface roughness are slightly different and difficult to control at the nanoscale. Similarly, controlling the vector-field and mode profile at the tip apex has remained challenging[6].

Hence, the ability to manipulate the excitation-field distribution at the apex consistently and in a deterministic fashion is highly desirable to provide for more reproducible TERS or TEPL enhancement, as well as engineer the polarization state of the tip-enhanced near-field. This precise control of the tip-localized surface plasmon (LSP) can extend the reach of near-field microscopy into a broader range of symmetry-selective nano-spectroscopy and -imaging applications, e.g., molecular orientation, excitons in van der Waals materials, quantum state selectivity, disorder, crystallinity, and ferroic order.

Recently, adaptive computational imaging approaches with deep learning algorithm have been permeating into almost all modalities of far-field optical imaging[7], ranging from astronomy to optimize spatial resolution and correct aberrations[8] to super-resolution biological imaging[9] and single atom imaging[10]. These approaches not only allow to develop novel optical instruments, e.g., for polarization control[11] and phase-resolved imaging[12], but also can establish a new field of research, e.g., adaptive optical imaging in turbid media[13]. However, despite these major benefits, these methods have not yet been applied for tip-enhanced near-field nano-spectroscopy and -imaging where they are expected to provide equal advances to the field.

In this work, we demonstrate modulating spatial coherence and polarization of the incident wavefront controlled by adaptive optics algorithms in tip-enhanced spectroscopy. In the implementation of adaptive TEPL (*a*-TEPL) and adaptive TERS (*a*-TERS), we achieve consistent improvement in field enhancement by optimizing the excitation wavefront for a given nanoscale morphology of the plasmonic tips. In addition, we demonstrate heterogeneous nano-optical responses from the same samples by manipulating the gradient field and near-field polarization dynamically.

## Results

### Tip-enhanced near-field spectroscopy with dynamical wavefront shaping.

Our experiment is based on a home-built tip-enhanced nano-spectroscope, with bottom-illumination and -detection as shown schematically in Fig. 1a. With wavefront shaping by a spatial light modulator (SLM), we can dynamically manipulate the tip-LSP characteristics, such as amplitude, phase, and polarization (see "Methods" for details). We achieve this tip near-field control based on the underlying mechanism of dynamical surface plasmon polariton (SPP) manipulation, as demonstrated recently for the plasmonic metal films[14–17]. Briefly, the SPP characteristics can be modulated depending heavily on the spatial coherence and wave-vector condition of the incident beam (see Supplementary Note 1 for details).

To determine the optimal near-field polarization state for a specific symmetry-selective TEPL or TERS signal, we design and operate a feedback loop based on a stepwise sequential algorithm[18]. Figure 1b shows the wavefront optimization process with spectrally integrated TEPL intensity for bright excitons of a WSe₂ monolayer. For wavefront shaping, we divide 600 × 600 pixels of the SLM active area into 12 × 12 segments, and control the temporal phase of each segment independently ranging from 0 to 2π. The algorithm starts with a random phase mask to eliminate the local optimum problem[18], even though the initial TEPL intensity is decreased compared to the signal with the flat wavefront. We then determine the optimal phase of the first segment providing the maximum TEPL intensity through an intensity feedback algorithm as a function of phase delay. This process is sequentially repeated for 144 (12 × 12) segments to find the optimized spatial phase mask, which provides the strongest TEPL signal (see Supplementary Note 2 for details). Note that the optimization sequence does not affect to the signal enhancement

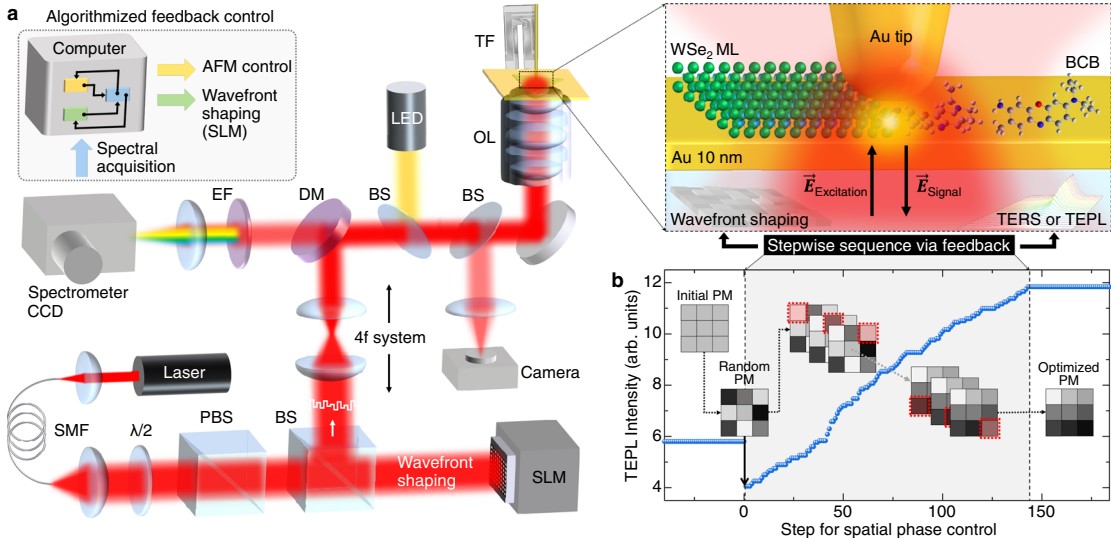

**Fig. 1 Adaptive nano-scope for vector-field controlled tip-enhanced nano-spectroscopy based on dynamic wavefront shaping. a** Schematic diagram of the experimental setup. For wavefront shaping, a He-Ne laser (λ = 632.8 nm) is spatially filtered and expanded to overfill the active area of spatial light modulator (SLM). The wavefront-shaped beam is imaged onto the back aperture of an objective lens (OL) in a 4f system for dynamical manipulation of the tip-LSP. Abbreviations: single-mode fiber (SMF), half-wave plate (λ/2), polarizing beam splitter (PBS), beam splitter (BS), dichroic mirror (DM), tuning fork (TF), and edge filter (EF). **b** Evolution of TEPL intensity of a WSe₂ monolayer during the optimization of spatial phase mask (PM) by a stepwise sequential algorithm. With the optimized PM, >2 × stronger TEPL response is obtained compared to conventional TEPL set-up. PMs in (**b**) are illustrations (see Supplementary Fig. 2 for real PMs we used).

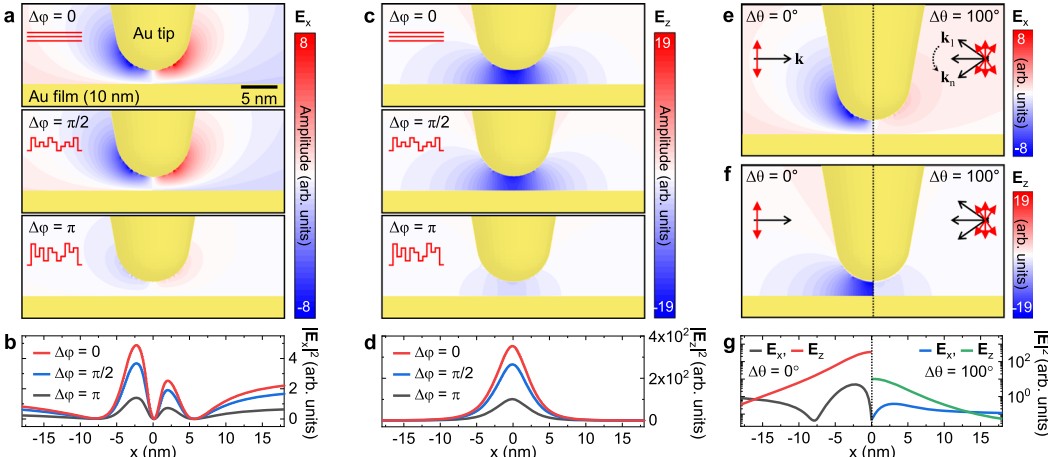

**Fig. 2 Finite-difference time domain (FDTD) analysis of wavefront shaping the tip apex field. a, c** Simulated in-plane ($\mathbf{E}_x$) and out-of-plane ($\mathbf{E}_z$) optical field distribution with respect to the spatial phase variation ($\Delta\varphi$, see Methods for details). **b, d** In-plane ($|\mathbf{E}_x|^2$) and out-of-plane ($|\mathbf{E}_z|^2$) optical field intensity profiles at the horizontal plane 1 nm underneath the tip derived from (**a, c**). **e, f** Comparison of $\mathbf{E}_x$ and $\mathbf{E}_z$ optical field distribution for optimal (left) and conventional (right) excitation polarization conditions by wave-vector variation ($\Delta\theta$, see "Methods" for details). **g** Optical intensity profiles at the horizontal plane 1 nm underneath the tip derived from (**e, f**), exhibiting significantly reduced excitation rate in the conventional experimental condition.

and it converges after completing a single cycle of the algorithm (see Supplementary Fig. 1 for details). It should be also noted that from the optimal phase masks obtained for different tips no systematic feature could be derived from the phase pattern. To systematically understand and control the near-field wavefront, a further careful study of characterizing the phase patterns is required.

**Simulated wavefront shaping effect in localized field enhancement.** To determine how the wavefront shaping leads to an optimized field enhancement, we calculate the optical field distribution in the vicinity of the Au tip for different excitation conditions using finite-difference time-domain (FDTD) simulations. Since the wavefront shaping can improve the spatial coherence, as well as the momentum mismatch by making an arbitrary polarization optimized for the Au tip, we perform the simulations separately for these two conditions (see "Methods" for details). Figure 2a, c shows the in-plane ($\mathbf{E}_x$) and out-of-plane ($\mathbf{E}_z$) optical field maps with respect to the phase delay ($\Delta\varphi$) of incident wavefronts ranging from 0 to $\pi$. As expected, both $\mathbf{E}_x$ and $\mathbf{E}_z$ field confinement becomes weak as the spatial coherence becomes worse. This result can be easily understood by the superposition principle of the waves. Figure 2b, d shows the in-plane ($|\mathbf{E}_x|^2$) and out-of-plane ($|\mathbf{E}_z|^2$) optical field intensity profiles at the horizontal plane 1 nm underneath the tip which indicate the excitation rate for TEPL and TERS experiments. Notably, the excitation rate for $\Delta\varphi = \pi$ is decreased less than half compared to the optimal condition ($\Delta\varphi = 0$).

To understand the effect of polarization, we then simulate the $\mathbf{E}_x$ and $\mathbf{E}_z$ distributions with an optical source having wave-vector variation $\Delta\theta = 100°$, as shown in Fig. 2e, f. The calculated field maps clearly show the significance of polarization (or momentum) matching in tip-enhanced nano-spectroscopy. Since we necessarily use focusing optics, e.g., objective lens and parabolic mirror[19,20], for the tip excitation, the simulated deterioration in the field enhancement is unavoidable. For example, we use an objective lens with NA = 0.8 and we estimate the wave-vector variation of ~100° for the excitation beam. In this case, we expect the excitation rate will be an order of magnitude lower than the ideal polarization condition, as can be seen in Fig. 2g. Therefore, the dynamic wavefront shaping with feedback of TEPL or TERS

signal can significantly reduce this loss by facilitating an arbitrary polarization optimized for the tip.

**Adaptive tip-enhanced PL and Raman responses of a WSe₂ monolayer.** Figure 3a shows a comparison of far-field PL (black) and TEPL spectra of a WSe₂ monolayer with conventional excitation (blue) and with wavefront shaping by the optimized spatial phase mask (a-TEPL, red). Since neutral excitons spread over the 2D crystal with fully in-plane electric dipole moment, we believe the in-plane polarization component of the tip-LSP is maximized through the wavefront shaping, in addition to the effect of the optimally shaped wavefront for the Au tip. Hence, we obtain highly sensitive near-field optical responses of 2D TMDs compared to previous conventional approaches[21–23]. Figure 3b shows the spectral evolution of a-TEPL response with respect to the distance between Au tip and the crystal on Au film. a-TEPL gradually increases from $d = 10$ nm to 5 nm owing to the localized antenna effect of Au tip, and is further enhanced at $d <$ 5 nm by the strong interaction between tip-dipole and image-dipole, which shows a similar behavior to the normal TEPL response[24,25]. We then obtain a-TEPL image of a WSe₂ monolayer with the same Au tip to quantify the spatial resolution, which is needed to estimate TEPL enhancement factor. a-TEPL image shows a heterogeneous response due to the Au substrate effect (Fig. 3c) and a spatial resolution of ~14 nm is confirmed, as shown in Fig. 3d. With the measured far-field and TEPL spectra (Fig. 3a), we calculate TEPL enhancement factor after compensating the area difference occupied by the laser focus and the tip near-field (see Supplementary Note 4 for details). We obtain ~$4.4 \times 10^4$-fold intensity enhancement with a-TEPL compared to far-field PL which is >2× larger than the normal TEPL intensity without wavefront shaping. Note that, in our experiment, far-field PL intensity is generally increased ~10% with the wavefront shaping compared to the normal excitation case. In the case of using a radially polarized excitation beam, we also observe a significant increase in a-TEPL signal compared to the TEPL signal without the SLM (see Supplementary Note 5 for details) showing that this effect is not dependent on type of excitation beam polarization. Besides, the optimized phase mask for tip-enhanced measurement differs from individual Au tip to the next. These features indicate that the augmented TEPL response is

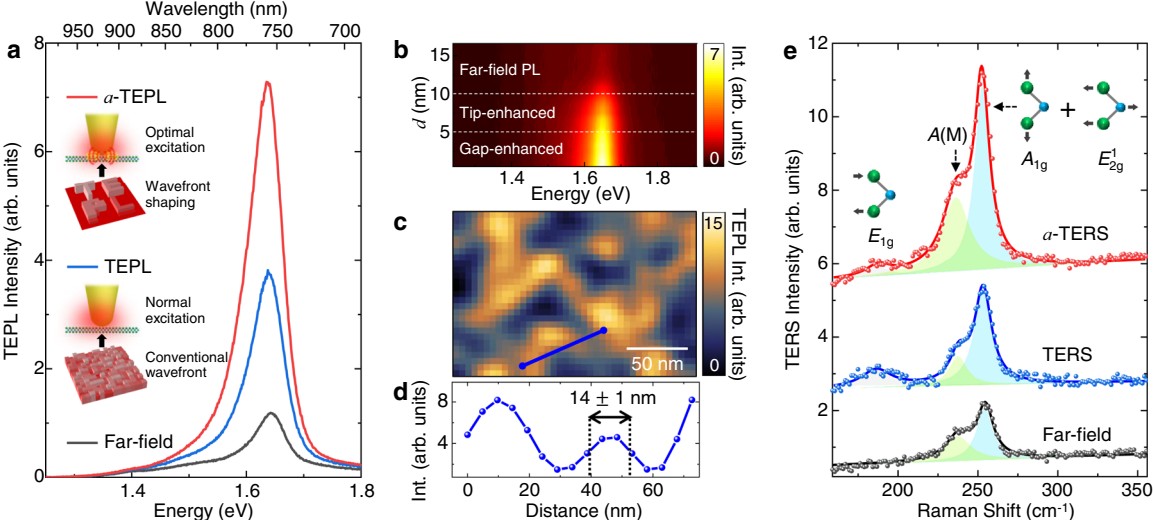

**Fig. 3 Augmented TEPL and TERS responses through the wavefront optimization process. a** Comparison of far-field PL (black) and TEPL spectra of a WSe₂ monolayer without (blue) and with wavefront shaping (*a*-TEPL, red). **b** Evolution of *a*-TEPL spectra as a function of distance *d* between Au tip and the crystal (WSe₂ monolayer), exhibiting gradual PL enhancement through the tip-enhanced and gap-enhanced regions. **c** *a*-TEPL image of a WSe₂ monolayer on Au film, and (**d**) *a*-TEPL intensity profile indicated as a blue line in Fig. 3c, showing a spatial resolution of ~14 nm. The same Au tip and sample were used for the results of Fig. 3a–d. **e** Comparison of far-field Raman (black) and TERS spectra of a WSe₂ monolayer without (blue) and with wavefront shaping (*a*-TERS, red) with Lorentz profile fit. Raman active $E_{1g}$, $A(M)$, and $A_{1g} + E_{2g}^1$ modes are observed. See SI for real PMs we used for the measurements (**a**) and (**e**).

mainly attributed to the tip-LSP control rather than the improved focusing quality of the excitation beam.

We then perform TERS experiments for the same sample to understand the wavefront shaping effect for Raman responses. Figure 3e shows far-field Raman (black) and TERS spectra with conventional excitation (blue) and with wavefront shaping by the optimized spatial phase mask (red). The observed peaks at ~187 cm⁻¹, ~238 cm⁻¹, and ~253 cm⁻¹ correspond to $E_{1g}$ (in-plane vibration of Se atoms), $A(M)$ (asymmetric phonon mode at the M point), and $A_{1g} + E_{2g}^1$ (degenerated mode for out-of-plane vibration of Se atoms and in plane vibration of W and Se atoms) modes[26,27]. The feedback process determining the optimal phase mask is performed for TERS intensity of the most prominent $A_{1g} + E_{2g}^1$ mode, and the obtained phase mask is totally different to the phase mask for *a*-TEPL measurement in Fig. 3a since the optimization target is different. Through the wavefront shaping (*a*-TERS), the peak intensity of $A_{1g} + E_{2g}^1$ mode is increased >2× compared to the normal TERS result, whereas TERS intensity of $E_{1g}$ mode (in-plane) is slightly decreased. This result indicates that the tip-LSP is optimized to maximize the out-of-plane vibrational mode ($A_{1g}$), in contrast to the near-field polarization for in-plane excitons (Fig. 3a). It should be noted that the adaptive tip-enhanced signal intensity was generally increased 1.3 ~ 2.5 × compared to normal TEPL or TERS intensity (without wavefront shaping) for the most tips we used (see Supplementary Note 6 for details). We guess that this variation in the additional enhancement rate is originated from the different symmetricity of the tip shape. Hence, although the enhancement rate via wavefront shaping can be different, we expect this approach works for both the electrochemically etched tips and the refined nano-fabricated tips, by focused ion beam (FIB) milling or field-directed sputter sharpening (FDSS)[28], in AFM-based and scanning tunneling microscopy (STM)-based TERS and TEPL spectroscopy. Note that, in our several control measurements, we find no significant improvement in spatial resolution in adaptive tip-enhanced imaging.

**Symmetry-selective tip-enhanced Raman scattering of BCB molecules**. To investigate more diverse spectroscopic responses with respect to the near-field polarization-control, we then measure heterogeneously oriented brilliant cresyl blue (BCB) molecules with *a*-TERS. While Raman scattering of the spin-coated molecules on Au film with a few monolayer coverages is not detected with far-field measurement, strong Raman signals emerge from *a*-TERS measurement with distinct spectral responses depending on the wavefront shaping condition, as shown in Fig. 4a. With the optimized excitation wavefront $\phi_1$, we observe *a*-TERS spectrum similar to normal TERS response, but with >2× enhanced scattering intensity. In contrast, we observe the other strong *a*-TERS peak at ~613 cm⁻¹ with the manipulated wavefront $\phi_2$. For the identification of the normal modes and symmetry properties of the observed *a*-TERS peaks, we calculate Raman-active (green) and IR-active (blue) vibrational spectra of a BCB molecule, as shown in Fig. 4b. The newly emerging peak at ~613 cm⁻¹ originates from out-of-plane C-C bends (strong, IR-active) and O-C₂ and N-C₂ twists (weak, Raman-active), in contrast to the in-plane stretching modes of ~583.5 cm⁻¹ (strong, Raman-active), as illustrated in Fig. 4c. Thus, the distinct *a*-TERS spectra are attributed to the symmetry-selective phonon-plasmon coupling by the near-field polarization-control. In addition, strong in-plane and out-of-plane near-field gradient (Fig. 4d) coupled to molecular quadrupole-quadrupole interaction[29] makes possible for IR-active modes to emerge in the visible spectral region (see Supplementary Note 7 for details). In contrast to the previous studies observing randomly appearing IR-active modes in the static plasmonic cavities[30–32], our approach provides dynamic controllability to turn IR-active modes on and off through the systematic phase modulation with <20 ms temporal resolution (see Supplementary Note 8 for details). Therefore, in addition to tip-enhanced nano-spectroscopy applications, our adaptive near-field approach can be more broadly used to control light-matter interactions in various plasmonic cavities[33–35].

From this *a*-TERS result, we can illustrate the inhomogeneous orientation of molecules with localized optical fields and field

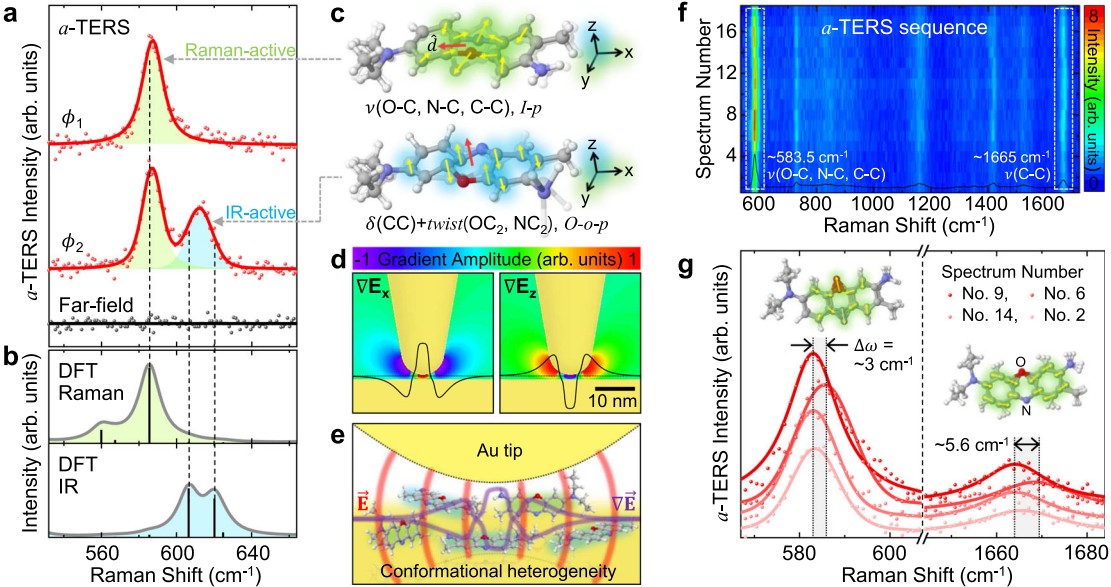

**Fig. 4 Symmetry-selective tip-enhanced Raman spectroscopy of BCB molecules. a** Measured far-field (black) and wavefront-shaped TERS (*a*-TERS, red circle) spectra for BCB molecules with Voigt profile fit (red line). Distinct *a*-TERS spectra ($\phi_1$ and $\phi_2$) are observed with near-field polarization-control. **b** Calculated vibrational spectra of a BCB molecule with Raman-active (green) and IR-active (blue) normal modes. **c** Illustrations for the vibrational modes of ~583.5 cm⁻¹ (top) and ~620 cm⁻¹ (bottom) peaks (vibrational mode of ~606 cm⁻¹ peak is similar to ~620 cm⁻¹ peak). Blue, red, gray, and white atoms indicate nitrogen, oxygen, carbon, and hydrogen, respectively. Abbreviations used: stretching ($\nu$), bending ($\delta$), in-plane (*i-p*), out-of-plane (*o-o-p*), and dipole derivative unit vector ($\hat{d}$). **d** Simulated in-plane ($\nabla E_x$) and out-of-plane ($\nabla E_z$) optical field gradient distribution in the vicinity of Au tip with gradient amplitude profile at the horizontal plane 1 nm underneath the tip (black line). Hyperbolic tangent function is used for normalized amplitude scale. **e** Illustration of the conformational heterogeneity of molecules in the tip-substrate gap. **f** Time-series of *a*-TERS spectra with dynamic wavefront shaping, exhibiting spectral fluctuation due to the symmetry-selective TERS response of heterogeneously oriented molecules. **g** Selected *a*-TERS spectra from the regions indicated by the white dashed boxes in (**f**). Voigt profile fits (solid line) show apparent peak shift of the in-plane stretching modes (~583.5 cm⁻¹ and ~1665 cm⁻¹).

gradient underneath the tip (Fig. 4e). The heterogeneity of BCB molecules are also confirmed with spectral fluctuations in *a*-TERS sequence because the varying near-field polarization during the feedback process allows for the selective excitation of the inhomogeneous molecules. Figure 4f shows a series of *a*-TERS spectra obtained in a specific time segment during the procedure of stepwise sequence, i.e., with changing spatial phase of the wavefront. As can be seen in Fig. 4f, g, in-plane stretching modes at ~583.5 cm⁻¹ and ~1665 cm⁻¹ show the most sensitive intensity fluctuation with TERS peak shift up to ~5.6 cm⁻¹ due to the significant intermolecular coupling to the metal substrate depending on the conformational states[36]. In other words, the observed *a*-TERS peak shifts originate from the different orientations of the selectively probing molecules because the near-field polarization at the tip apex can be changed during the wavefront shaping (see Supplementary Note 9 for more details). This experiment demonstrates that probing conformational heterogeneity of a few molecules is feasible even at room temperature with high sensitivity and specificity of *a*-TERS. The inverse can also be done, with these results certifying the ability of dynamic control of the tip-LSP through the wavefront shaping. Note that, for the tip-induced polarization for vibrational modes of a sample in *a*-TERS[37], the Raman tensor and the tip tensor are not influenced by the wavefront shaping and only the incident light can be changed.

## Discussion

In conclusion, our approach of adaptive tip-enhanced nano-spectroscopy and -imaging provides a versatile method to enhance field localization and control of the vector near-field of

plasmonic tips. By exploiting a computational algorithm to overcome the structural limitations of tip-engineering by near-field shaping, adaptive nano-imaging provides a generalizable solution for reliable field enhancement and more consistent data quality advancing from single-molecule TERS[3–5] to single emitter tip-enhanced strong coupling (TESC) experiments[1,2] with fast modulation and control of the tip-LSP polarization. In addition, it will facilitate a broader range of symmetry-selective and phase-resolved nano-spectroscopy and -imaging. Furthermore, when this spatial coherence control is combined with ultrafast coherent control methods[38–41], it can greatly enhance optical sensitivity of ultrafast nano-spectroscopy, nonlinear and coherent nano-imaging, and nano-optical crystallography.

## Methods

**FDTD simulations of optical field distribution**. We used a commercial finite-difference time-domain (FDTD) simulation software (Lumerical Solutions, Inc.) to characterize the optical field distribution at the Au tip-Au film nano-gap. An Au tip with a 5 nm apex radius was placed in close proximity (2 nm gap) to a 10 nm thick Au film. As a fundamental excitation source, a monochromatic 633 nm light was used with a linear polarization in parallel with respect to the tip axis. To characterize the local field enhancement with respect to the spatial phase difference, 25 optical sources were used with different phase delay $\Delta\varphi$, as shown in Fig. 2a–d. In these simulations, we designed the total field amplitude of 1 for 25 optical sources with the same linear polarization with different phase delay ranging from 0 to $\pi/2$ for Fig. 2a (middle) and from 0 to $\pi$ for Fig. 2a (bottom). Wave-vector variation of incident wavefronts ($\Delta\theta$), is another important parameter to increase the field enhancement. In conventional TERS or TEPL set-up, the focusing optics, e.g., objective lens or parabolic mirror, have a large degree of wave-vector variation. To understand the effect of wave-vector to the local field enhancement, 25 optical sources were used with different propagation direction ($\Delta\theta = 100°$) but the same phase delay ($\Delta\varphi = 0$), as shown in Fig. 2e–g.

**Sample preparation**. An Au film was used as the sample substrate for the bottom-illumination mode TEPL and TERS experiments. The gold film was deposited onto a coverslip with 10 nm thickness using a thermal evaporator. Note that, before the Au-deposition, the cleaning of coverslips was done by ultrasonication with acetone, isopropanol, and ethanol for 15 min each. To transfer the chemical vapor deposition (CVD)-grown $WSe_2$ monolayers onto the flat Au film/coverslip, a wet transfer process was used. As a first step, poly(methyl methacrylate) (PMMA) was spin-coated onto $WSe_2$ monolayers grown on the $SiO_2$ substrate. The PMMA coated $WSe_2$ monolayers were then separated from the $SiO_2$ substrate using a hydrogen fluoride solution, and carefully transferred onto the Au film/coverslip. The transferred crystals on Au film was then rinsed in distilled water to remove residual etchant, and dried naturally for 6 h to improve the adhesion. Last, the PMMA was removed using acetone. For molecular TERS experiment, the brilliant cresyl blue (BCB) sample was prepared on the Au film/coverslip by spin coating (3000 rpm for 15 s) from an ethanol solution with a few monolayer coverages. The BCB sample was then covered by $Al_2O_3$ capping layer (0.5 nm thickness) using atomic layer deposition (ALD) to suppress thermally activated processes of molecules at room temperature, such as rotational and structural diffusions[20].

**TEPL and TERS setup**. We built a tip-enhanced nano-spectroscopy setup with the bottom-illumination mode for measuring TEPL and TERS responses, as shown in Fig. 1a. A continuous wave helium-neon laser ($\lambda = 632.8$ nm, ≤0.6 mW) was spatially filtered using a single mode fiber (core diameter of ≤3.5 μm) and expanded to a 25 mm diameter beam to improve spatial coherence. Using a combination of a half wave plate ($\lambda/2$) and a polarizing beam splitter (PBS), only the horizontally polarized beam was sent to a phase-only SLM (LCSO-SLM, Hamamatsu). The wavefront-shaped beam reflected from the active area of a SLM was imaged onto the back aperture of an objective lens (0.8 NA, Olympus) in a 4f system, and focused into the junction between the electrochemically etched Au tip and sample. To regulate the horizontal and vertical position of the Au tip with 0.2 nm precision, a shear-force atomic force microscopy (AFM) was used under ambient conditions. Tip-enhanced spectroscopic signals were collected using the same objective lens and detected using a spectrometer (f = 328 mm, Kymera 328i, Andor) with a thermoelectrically cooled charge-coupled device (CCD, DU420A, Andor), after passing through an edge filter (633 nm cut-off). The spectrometer was calibrated using a mercury lamp, and the spectral resolutions were characterized with ~1.6 nm in wavelength and 39.8 $cm^{-1}$ in wavenumber for a 150 g/mm grating (used for TEPL experiments) and 0.3 nm in wavelength and ~7 $cm^{-1}$ in wavenumber for a 1200 g/mm grating (used for TERS experiments).

**DFT calculations of vibrational modes**. To obtain theoretical normal modes and their vibrational amplitude of the BCB molecule, density functional theory (DFT) calculations were performed using a commercial Gaussian09 program. The ground-state geometry of the BCB molecule was initially optimized at the Becke three-parameter Lee-Yang-Parr (B3LYP), and the optimal vibrational frequency ($\omega$) for the geometry was obtained with the basis set of 6-311++G(d,p). The calculated Raman and IR spectra were plotted by taking the spectral resolution of the experiment (~7 $cm^{-1}$) into account, as shown in Fig. 4b.

## Data availability

The data that support the plots within this paper and other findings of this study are available from the corresponding author upon reasonable request.

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

## Acknowledgements

We thank Jung-Hoon Park and Hyuk Kyu Pak for insightful discussions. This work was supported by the 2018 Research Fund (1.180091.01) of UNIST(Ulsan National Institute of Science & Technology) and the National Research Foundation of Korea (NRF) grant funded by the Korea government (MEST) (No. 2019R1F1A1059892, 2019K2A9A1A06099937, and 2020R1C1C1011301). M.B.R. thanks the National Science Foundation (NSF Grant CHE 1709822) for support. M.S.J. thanks Creative Materials Discovery Program (NRF-2019M3D1A1078304) funded by the Ministry of Science and ICT.

## Author contributions

K.-D.P. and D.Y.L. conceived the experiment. C.P., M.K., and M.S.J. designed and prepared the samples. K.-D.P., Y.K., and D.Y.L. performed the measurements. K.-D.P. and J.C. performed the simulations. K.-D.P., M.B.R., and D.Y.L. analysed the data, and all authors discussed the results. K.-D.P., M.B.R., and D.Y.L. wrote the manuscript with contributions from all authors. K.-D.P. supervised the project.

## Competing interests

The authors declare no competing interests.
