## [Peer Review File · Nature Communications]

REVIEWER COMMENTS

Reviewer #1 (Remarks to the Author):

The authors present experimental results of tip-enhanced photoluminescence (TEPL) and tip-enhanced Raman scattering (TERS) spectroscopies with phase shaped structured illumination. They apply this new technique to probing conformation of single BCB molecules and 2D monolayer WSe₂ materials. They claim a factor of 2 TEPL signal enhancement of WSe₂ and controlled observation of IR-active modes in TERS of BCB. These new and interesting results may be published after addressing the following comments:

- 1) The title is confusing: it is not clear what means "adaptive" in this work. It seems that the authors just scan the parameters of the shaped wave front and optimize pixel by pixel sequentially. This is not an adaptive approach but rather a deterministic sequential parameter optimization. Therefore, it would be more appropriate to call it "Optimal" instead of "Adaptive." Correspondingly, abbreviations a-TEPL and a-TERS are not appropriate.
- 2) Does the optimization sequence matter? Did the authors try to optimize the same target by using several different sequences? Did they give similar results? And why?
- 3) Can one learn anything from the optimal phase patterns?
- 4) In the abstract, the authors claim "irreproducible ... imaging" and addressing the "inverse" problem. However, in the manuscript these two claims are not clearly addressed. They should be either removed from the abstract or addressed. For example, the tip-fabrication imperfection leads to variable results from tip to tip. That is correct. The authors make a claim in the abstract that they solved that problem by "inverse approach" by shaping the incident field. That is not correct because even though the authors optimized the field-tip coupling by shaping there are still tip to tip variations due to tip-fabrication imperfections. The authors did not show that they can get similar results for several different tips. Therefore this claim is not supported.

Reviewer #2 (Remarks to the Author):

The paper entitled "Adaptive tip-enhanced nano-spectroscopy" by Kyoung-Duck Park et. al. is devoting to further developing TERS and TERL techniques through adaptive optics. By shaping a dynamical wavefront of the excitation with a SLM, the authors succeeded to improve the reproducibility of a TERS/TERL response due to a computational algorithm. In a sense, a tip itself serves a simple spatial modulator, that allows one to enhance a signal and to overcome the diffraction limit. In this paper, a focus is shifted into the spatial modulation of the excitation, in addition providing a robustness and reproducibility of an optical response. This manuscript is well organized and all conclusions are clear. This study contributes into the cutting-edge field of calculation-assisted TERS/TERL and may be of keen interest to the broad Readership and can be potentially recommended for publication in Nature Communications after making major revision.

1. The authors address the issue concerning a challenge of controlling the near fields beneath the tip apex beneath because of irreproducibility of its size, shape and roughness. Probably, it concerns ways to produce the tip, this is the case for electrochemical etching rather than, for example, focused ion (Ar⁺) beam milling suggested by A. Apkarian et. al. ACS NANO (2017). What kind of technique was used for fabricating a tip? Besides, a successful demonstration of a sub-nanometer resolution with STM-based TERS in ultrahigh vacuum at cryogenic temperatures shows quite reproducible results (Dong et. al. Nature (2013), Apkarian et. al. Nature (2019)). The developed approach is of interest when a tip is driven by atomic-force or shear-force microscopy, right?

2. page 3, it is not clear what the authors mean under saying "...tips often do not uniform local field enhancement...", in TERS it is played a role by only the external electric field that is commonly non-uniform, unlike the inner electric field for nanostructures enough small compared to a light wavelength.
3. Why the authors say about surface plasmon polaritons rather than localized plasmons or gap modes?
4. How it is taken into account information on symmetry of vibrational modes (Raman tensors), a tip (tip tensor) and a substrate (substrate tensor). Whether the depolarization effects are considered?
5. The authors used an objective lens with NA=0.8 in the experiment whereas they use a high NA objective, that is, NA>1 in the supplementary information. Why the authors did not utilize the immersion objective with a higher NA to squeeze the electric field in the best way.
6. page 13, the following sentence "The spectrometer was calibrated using a mercury lamp, and the spectral resolutions were characterized with 1.6 nm for a 150 g/mm⁻¹ grating (for TEPL experiments) and 7 cm⁻¹ for a 1200 g/mm⁻¹ grating (for TERS experiments)." looks strange, the authors get mixed up spatial resolution and spectral resolution, in particular, a 1200 grooves per mm grating gives ca. 1.3 cm⁻¹ for a 633 nm wavelength, what does 150 g/mm⁻¹ and 1200 g/mm⁻¹ mean?
7. The authors say nothing about the spatial coherence of optical near fields, but they make attempts to improve it through its modulating with a SLM.
8. In Fig. 1 (a) a radial convertor is not shown. How wavefront shaping affects the radial polarization state. Whether it is needed to commonly use the radial convertor in the experiment?

Reviewer #3 (Remarks to the Author):

This work describes the first application of adaptive optics and imaging concepts to near-field spectroscopy and imaging. It represents an important advance in the field, and is likely to be adopted and further developed by many groups. In demonstrating its application to tip-enhanced PL and Raman of 2D semiconductors (TMDs) and molecules, the authors highlight signal enhancements ($\sim 2\times$) over standard illumination schemes and, perhaps more excitingly, symmetry selective phonon-plasmon coupling – a big advantage of this new development, with implications for control of light-matter interactions in the quantum regime. The authors have performed key control experiments, including the check for convergence of their optimization algorithm after a single cycle. However, there are several points and issues which must be addressed (given below) before I believe this manuscript would be suitable for publication in Nature Communications.

1) By describing their work as the demonstration of engineering "a robust nano-optical response with full polarization and gradient field control", the authors seem to imply that the approach enables full, arbitrary vector-field engineering of the near-field surrounding a nano-optical probe. But in fact the situation is a more nuanced one – and much more constrained. While there is certainly some crucial control, it is not completely flexible/arbitrary. Rather, every tip is different at the nanoscale, leading to nanoscale variations in polarization response and SPP field distribution from tip to tip. More specifically, engineering the vector field state is constrained by uncontrollable nanoscale morphology and texturing at the tip apex. Instead of full arbitrary control over each tip's near-field, this adaptive approach is more correctly described as optimizing the field properties given a specific (and heterogeneous) nanoscale morphology (e.g., different nanoscale textures/roughness for different tips).

The authors are requested to change the discussion/description of the technique to emphasize the concept that the method is best suited for maximizing overlap of nanoscale optical response of a tip, which varies from tip to tip, with the response of the sample property one is interested in probing.

2) the manuscript generally seems to be missing critical references and is light in acknowledging previous work in the field, particularly in regards to the near-field study of TMDs. For example, this includes the first near-field imaging and spectroscopy investigations of TMDs [Lee, et al., *Nanoscale* 7, 11909 (2015); Bao, et al., *Nature Communications* 6, 7993 (2015)], recent studies of metal-specific effects on gap-mode TERS of WSe₂ [Krayev, et al., *J. Phys. Chem. C* 124, 8971 (2020)], and nano-quantum-optical control of TMD emission [He, et al., *Science Advances* 5, eaau8763 (2019)].

3) At the bottom of page 6, the statement "Since excitons spread..." should be updated to "Since free excitons spread...with fully in-plane electric dipole moment", since strain- and defect-localized excitons can have notable out-of-plane character (Luo, et al., *Nano letters* 20, 5119 (2020)).

4) In Fig. 3b, there appears to be no quenching at $d = 0$ (tip in contact with the WSe₂). It is well known that radiation from emitters is usually quenched when in contact with metal surfaces (due to coupling to surface bound traveling plasmon-polaritons, electron-hole excitation in the metal (followed by non-radiative recombination, etc)). What do the authors attribute this lack of quenching to?

5) In the 2nd sentence on pg. 8, the authors state: "...as well as the largest plasmon-enhanced PL intensity of a transition metal dichalcogenide (TMD) monolayer reported to date [22]." This seems unprovable, and likely not true. For example, strain localized emitters with out-of-plane dipoles are likely enhanced by a larger factor, due to the mode polarization profile of tip-substrate gap mode. The authors should remove this claim from the text and the abstract.

Point-by-point response:

Reviewer #1:

The authors present experimental results of tip-enhanced photoluminescence (TEPL) and tip-enhanced Raman scattering (TERS) spectroscopies with phase shaped structured illumination. They apply this new technique to probing conformation of single BCB molecules and 2D monolayer WSe₂ materials. They claim a factor of 2 TEPL signal enhancement of WSe₂ and controlled observation of IR-active modes in TERS of BCB. These new and interesting results may be published after addressing the following comments:

We thank the reviewer for the appreciation of the novelty of our approach. With regard to the technical comments below, we have made corresponding revisions to our manuscript as indicated in red in the revised manuscript file.

1) The title is confusing: it is not clear what means “adaptive” in this work. It seems that the authors just scan the parameters of the shaped wave front and optimize pixel by pixel sequentially. This is not an adaptive approach but rather a deterministic sequential parameter optimization. Therefore, it would be more appropriate to call it “Optimal” instead of “Adaptive.” Correspondingly, abbreviations a-TEPL and a-TERS are not appropriate.

We thank the reviewer for the helpful comment, but we have slightly different idea for this comment. In “adaptive optics,” many different algorithms are used to find, e.g., an optimal wavefront iteratively. In our work, we use a deterministic sequential parameter optimization to find the optimal phase mask which gives the largest TEPL or TERS signal (Fig. 3). In addition, as demonstrated in Fig. 4, we manipulate the wavefront adaptively to measure specific molecules or Raman modes. Indeed, we are trying many different algorithms, including both sequence and genetic algorithms, to expand the field of adaptive tip-enhanced nano-spectroscopy and -imaging, as follow up studies. Since this work introduces adaptive optics for tip-enhanced nano-spectroscopy for the first time and there will be many different adaptive optics algorithms to be applied, we believe the generalizable title adaptive tip-enhanced nano-spectroscopy is appropriate.

As a similar example, a previous work entitled “Adaptive wavefront shaping for controlling nonlinear multimode interactions in optical fibers” [Tzang et al., *Nat. Photon.* **12**, 368 (2018)] just used a simple genetic algorithm for deterministic signal optimization [Vellekoop et al., *Opt. Express* **23**, 12189 (2015)]. Hence, we believe the terminologies of a-TEPL and a-TERS are quite reasonable to indicate this generalizable approach in future works of us and other followers.

2) Does the optimization sequence matter? Did the authors try to optimize the same target by

using several different sequences? Did they give similar results? And why?

From our experiments, the optimization sequence does not matter. To verify this property, we performed the sequence algorithm with different initial random phase patterns. Using the same target signal, we obtained the same signal enhancement regardless of the initial phase patterns.

Additionally, to confirm the convergence, we repeated the algorithm three times continuously with the target signal. With several control experiments with different tips, we could find that the saturation enhancement was obtained and the optimization was converged after completing a single cycle of the algorithm, as shown in Fig. S1. From these results, we think that the optimal phase of each element in the wavefront is independently determined and not affected by the neighbor elements. A previous study [Vellekoop et al., *Opt. Express* **23**, 12189 (2015)] also mentioned that “Regardless of what algorithm is used, ideally all methods converge to the unique global optimum.”

We have added a note to the manuscript to emphasize this point as follows:

[Added text] Note that the optimization sequence does not affect to the signal enhancement and it converges after completing a single cycle of the algorithm (see SI for details).

3) Can one learn anything from the optimal phase patterns?

Unfortunately, we could not obtain any useful information from the optimal phase patterns, such as properties for the tip structure or excitation polarization. First, the optimized phase mask to obtain the strongest TEPL/TERS signal differs tip-to-tip because the nano-structure of tip apex is always slightly different. From the obtained optimal phase masks from different tips, we tried to find regularities but could not find any systematic feature. Nevertheless, we believe that to systematically understand and control the near-field wavefront would be a meaningful goal for future studies, e.g., to perform control experiments with more plasmonic tips by characterizing the optimal phase patterns as well as the angular radiation patterns from the tip [Bohmler et al., *Opt. Express* **18**, 16443 (2010)].

To provide clear information to readers, we have noted a brief discussion in the revised manuscript as follows:

[Added text] It should be also noted that from the optimal phase masks obtained for different tips no systematic feature could be derived from the phase pattern. To systematically understand and control the near-field wavefront, a further careful study of characterizing the phase patterns is required.

4) In the abstract, the authors claim “irreproducible ... imaging” and addressing the “inverse” problem. However, in the manuscript these two claims are not clearly addressed. They should

be either removed from the abstract or addressed. For example, the tip-fabrication imperfection leads to variable results from tip to tip. That is correct. The authors make a claim in the abstract that they solved that problem by “inverse approach” by shaping the incident field. That is not correct because even though the authors optimized the field-tip coupling by shaping there are still tip to tip variations due to tip-fabrication imperfections. The authors did not show that they can get similar results for several different tips. Therefore this claim is not supported.

We agree and to avoid this confusion we have revised the sentences in the Abstract as follows:

[Revised text] However, the techniques suffer from inconsistent signal enhancement and difficulty in polarization resolved measurement due to lack of precise nanoscale control of the tip apex geometry. To address this problem, we present adaptive tip-enhanced nano-spectroscopy approach optimizing the nano-optical vector-field at the tip apex via adaptive optics.

Reviewer #2:

The paper entitled “Adaptive tip-enhanced nano-spectroscopy” by Kyoung-Duck Park et. al. is devoting to further developing TERS and TERL techniques through adaptive optics. By shaping a dynamical wavefront of the excitation with a SLM, the authors succeeded to improve the reproducibility of a TERS/TERL response due to a computational algorithm. In a sense, a tip itself serves a simple spatial modulator, that allows one to enhance a signal and to overcome the diffraction limit. In this paper, a focus is shifted into the spatial modulation of the excitation, in addition providing a robustness and reproducibility of an optical response. This manuscript is well organized and all conclusions are clear. This study contributes into the cutting-edge field of calculation-assisted TERS/TERL and may be of keen interest to the broad Readership and can be potentially recommended for publication in Nature Communications after making major revision.

We thank the reviewer for acknowledging the novelty and significance of our work. We have made corresponding revisions to our manuscript in response to the comments made below as indicated in red.

1. The authors address the issue concerning a challenge of controlling the near fields beneath the tip apex beneath because of irreproducibility of its size, shape and roughness. Probably, it concerns ways to produce the tip, this is the case for electrochemical etching rather than, for example, focused ion (Ar+) beam milling suggested by A. Apkarian et. al. ACS NANO (2017). What kind of technique was used for fabricating a tip? Besides, a successful demonstration of a sub-nanometer resolution with STM-based TERS in ultrahigh vacuum at cryogenic temperatures shows quite reproducible results (Dong et. al. Nature (2013), Apkarian et. al. Nature (2019)). The developed approach is of interest when a tip is driven by atomic-

force or shear-force microscopy, right?

First, we guess that our wavefront shaping approach works better for the electrochemically etched tips (what we used) compared to the refined nano-fabricated tips by FIB milling or field-directed sputter sharpening (FDSS). Because the etched tips are less reproducible in size, shape, and surface roughness. But, a-TEPL and a-TERS will also work for the refined symmetric shape tips in AFM-based TERS/TEPL as well as STM-based TERS/TEPL because the optimal wavefront via adaptive optics can optimize the optical near-field at the apex of plasmonic tips. To summarize, we can expect additional signal enhancement compared to the normal TERS/TEPL through the wavefront shaping even though the additional enhancement rate can be different depending on the tip condition.

We have made notes for these features with citation of [N. Tallarida et al., *ACS Nano* **11**, 11393 (2017)] in the revised text as follows:

[Added text] It should be noted that the adaptive tip-enhanced signal intensity was generally increased 1.3 ~ 2.5× compared to normal TEPL or TERS intensity (without wavefront shaping) for the most tips we used (see SI for details). We guess that this variation in the additional enhancement rate is originated from the different symmetricity of the tip shape. Hence, although the enhancement rate via warfront shaping can be different, we expect this approach works for both the electrochemically etched tips and the refined nano-fabricated tips, by focused ion beam (FIB) milling or field-directed sputter sharpening (FDSS) [*ACS Nano* **11**, 11393 (2017)], in AFM-based and STM-based TERS and TEPL spectroscopy.

In this paragraph, it should be noted that we have changed the adaptive enhancement rate of “1.5 ~ 2.5×” into “1.3 ~ 2.5×” in the revised manuscript. Because our further experiments confirmed that the additional signal enhancement by the wavefront shaping is ~130 % for some Au tips.

2. page 3, it is not clear what the authors mean under saying “...tips often do not uniform local field enhancement...”, in TERS it is played a role by only the external electric field that is commonly non-uniform, unlike the inner electric field for nanostructures enough small compared to a light wavelength.

In the sentence of “Even under the same illumination conditions, the plasmonic tips often do not show uniform local field enhancement and the enhancement differs from tip to tip, since the apex size, shape, and surface roughness are difficult to control at the nanoscale,” we mean TERS/TEPL enhancement factor is non-uniform for different tips even though the excitation condition is the same because the slightly different apex conditions (shape, size, and roughness) of plasmonic tips give rise to non-uniform local field enhancement at the tip apex. To avoid confusion, we have revised the sentence as follows:

[Revised text] Even under the same excitation conditions, TERS or TEPL enhancement factor

is non-uniform for different plasmonic tips since their apex size, shape, and surface roughness are slightly different and difficult to control at the nanoscale.

3. Why the authors say about surface plasmon polaritons rather than localized plasmons or gap modes?

Indeed, the propagating SPP along with the tip affect to TEPL [*Opt. Exp.* **15**, 12131 (2007)], however, the main contribution is the localized SPP in our experiment. To avoid confusion, we have used a more general terminology; “localized surface plasmon (LSP)” in the revised text.

4. How it is taken into account information on symmetry of vibrational modes (Raman tensors), a tip (tip tensor) and a substrate (substrate tensor). Whether the depolarization effects are considered?

Taking into account the tip-enhancement, the induced polarization for vibrational modes of a sample is written as $P_{j,n} \propto F_v^{scat} \chi_{j,k,n} F_u^{inc} E_k^{inc}$ ($j, k = x, y, z$). Where F_u^{inc} and F_v^{scat} are the field enhancement factors of the incident and scattered light (u and v denote the polarization state) and $\chi_{x,y,z,n}$ and E_k^{inc} are the Raman tensor and electric field of the incident light. The field enhancement factors for the tip can be contracted approximately into a single tensor $F_{vu}^{TERS} = F_v^{scat} F_u^{inc}$ and the tip-enhanced Raman intensity is given by $I_{scat} \propto |F_{vu}^{TERS} \chi_{j,k,n} E_k^{inc}|^2$ [*J. Raman Spectrosc.* **40**, 1413 (2009)]. In *a*-TERS, this general formula is applied, but E_k^{inc} can be changed depending on the wavefront shaping condition.

Depolarization does not happen in static configurations, the light when interacting with even complex scattering geometries would polarization rotate and with some retardation a more complex polarization state can result, but not depolarization.

We thank the reviewer for pointing this out and we have added a note below in the revised manuscript.

[Added text] Note that, for the tip-induced polarization for vibrational modes of a sample in *a*-TERS [*J. Raman Spectrosc.* **40**, 1413 (2009)], the Raman tensor and the tip tensor are not influenced by the wavefront shaping and only the incident light can be changed.

5. The authors used an objective lens with NA=0.8 in the experiment whereas they use a high NA objective, that is, NA>1 in the supplementary information. Why the authors did not utilize the immersion objective with a higher NA to squeeze the electric field in the best way.

We think there was a misunderstanding by the reviewer. We did not use a high NA objective lens in this work. Currently, we are using an oil-immersion lens (NA = 1.3) as the reviewer also suggested, but we did not equip the oil-immersion lens in the lab in the stage of this work.

Note that we have confirmed the wavefront shaping effect (additional TERS/TEPL enhancement) is the same regardless of the NA of objective lenses.

6. page 13, the following sentence “The spectrometer was calibrated using a mercury lamp, and the spectral resolutions were characterized with 1.6 nm for a 150 g/mm-1 grating (for TEPL experiments) and 7 cm-1 for a 1200 g/mm-1 grating (for TERS experiments).” looks strange, the authors get mixed up spatial resolution and spectral resolution, in particular, a 1200 grooves per mm grating gives ca. 1.3 cm-1 for a 633 nm wavelength, what does 150 g/mm-1 and 1200 g/mm-1 mean?

The spectral resolution is determined by the focused beam size at the entrance slit of the spectrometer as well as the number of grooves per mm in the diffraction grating. From our direct characterization of the spectral resolution for the 150 g/mm and 1200 g/mm gratings, we obtained the results of 1.6 nm for a 150 g/mm grating and 7 cm⁻¹ for a 1200 g/mm grating. For TEPL and TERS measurements, we used the unit of spectral resolution in wavelength and wavenumber, respectively. We further correct a mistake in the unit of the grating (g/mm is correct). To avoid confusion, we have revised the sentences in the revised manuscript as follows:

[Revised text] The spectrometer was calibrated using a mercury lamp, and the spectral resolutions were characterized with 1.6 nm in wavelength and 39.8 cm⁻¹ in wavenumber for a 150 g/mm grating (used for TEPL experiments) and 0.3 nm in wavelength and 7 cm⁻¹ in wavenumber for a 1200 g/mm grating (used for TERS experiments).

7. The authors say nothing about the spatial coherence of optical near fields, but they make attempts to improve it through its modulating with a SLM.

In this work, we made attempts to improve or modify the spatial coherence of the “far-field” excitation beam via wavefront shaping to more effectively excite the plasmonic tip. In general, the “spatial coherence of near-field” is defined for the scattered light from the sub-wavelength structure [*Phys. Rev. Lett.* **113**, 186101 (2014), *Phys. Rev. A* **102**, 053509 (2020)]. In the case of plasmonic tips, the spatial coherence of near-field is low because the scatterer (tip) has an asymmetric geometry and nanoscale morphology. For example, a previous study demonstrated that a correlation length of ~30 nm for the optical phonons in graphene by TERS [*Phys. Rev. Lett.* **113**, 186101 (2014)]. We did not characterize the spatial coherence of TEPL and TERS signals in this study, but we expect that the near-field coherence is not affected by the wavefront shaping of the excitation field because we think the tip-scatterer can act as a point light source regardless of the excitation beam condition.

8. In Fig. 1 (a) a radial convertor is not shown. How wavefront shaping affects the radial

polarization state. Whether it is needed to commonly use the radial convertor in the experiment?

Since we used a combination of the linearly polarized light and the wavefront shaping by SLM for the results in the main text, we did not include a radial polarizer in Fig. 1(a). Indeed, the wavefront shaping by the SLM allows to convert the linear polarization state to any kind of polarization state including the radial polarization state [Bashkansky et al., *Opt. Express* **18**, 212 (2010)]. Although we demonstrated the effective wavefront shaping effect for the initially radially polarized excitation laser (Supplementary Information, Fig. S5) for control experiment, indeed we do not need a radial polarizer because the SLM can dynamically manipulate the polarization state.

Reviewer #3:

This work describes the first application of adaptive optics and imaging concepts to near-field spectroscopy and imaging. It represents an important advance in the field, and is likely to be adopted and further developed by many groups. In demonstrating its application to tip-enhanced PL and Raman of 2D semiconductors (TMDs) and molecules, the authors highlight signal enhancements (~2x) over standard illumination schemes and, perhaps more excitingly, symmetry selective phonon-plasmon coupling – a big advantage of this new development, with implications for control of light-matter interactions in the quantum regime. The authors have performed key control experiments, including the check for convergence of their optimization algorithm after a single cycle. However, there are several points and issues which must be addressed (given below) before I believe this manuscript would be suitable for publication in Nature Communications.

We thank the reviewer for recognizing the novelty of our work and providing helpful comments. With regard to the raised concerns, we have addressed them in the point-by-point response below. We have also made corresponding revisions to our manuscript as indicated in red.

1) By describing their work as the demonstration of engineering “a robust nano-optical response with full polarization and gradient field control”, the authors seem to imply that the approach enables full, arbitrary vector-field engineering of the near-field surrounding a nano-optical probe. But in fact the situation is a more nuanced one – and much more constrained. While there is certainly some crucial control, it is not completely flexible/arbitrary. Rather, every tip is different at the nanoscale, leading to nanoscale variations in polarization response and SPP field distribution from tip to tip. More specifically, engineering the vector field state is constrained by uncontrollable nanoscale morphology and texturing at the tip apex. Instead of full arbitrary control over each tip’s near-field, this adaptive approach is more correctly described as optimizing the field properties given a specific (and heterogeneous) nanoscale morphology (e.g., different nanoscale textures/roughness for different tips). The authors are requested to change the discussion/description of the technique to emphasize the concept

that the method is best suited for maximizing overlap of nanoscale optical response of a tip, which varies from tip to tip, with the response of the sample property one is interested in probing.

We agree with the reviewer's opinion and thank for the helpful suggestion. To avoid overemphasis of our work, we have revised the sentences in the Introduction as follows:

[Revised text] In the implementation of adaptive TEPL (*a*-TEPL) and adaptive TERS (*a*-TERS), we achieve consistent improvement in field enhancement by optimizing the excitation wavefront for a given nanoscale morphology of the plasmonic tips. In addition, we demonstrate heterogeneous nano-optical responses from the same samples by manipulating the gradient field and near-field polarization dynamically.

2) the manuscript generally seems to be missing critical references and is light in acknowledging previous work in the field, particularly in regards to the near-field study of TMDs. For example, this includes the first near-field imaging and spectroscopy investigations of TMDs [Lee, et al., *Nanoscale* 7, 11909 (2015); Bao, et al., *Nature Communications* 6, 7993 (2015)], recent studies of metal-specific effects on gap-mode TERS of WSe₂ [Krayev, et al., *J. Phys. Chem. C* 124, 8971 (2020)], and nano-quantum-optical control of TMD emission [He, et al., *Science Advances* 5, eaau8763 (2019)].

We thank the reviewer for the suggested references which we have included in the manuscript.

3) At the bottom of page 6, the statement "Since excitons spread..." should be updated to "Since free excitons spread...with fully in-plane electric dipole moment", since strain- and defect-localized excitons can have notable out-of-plane character (Luo, et al., *Nano letters* 20, 5119 (2020)).

We thank the reviewer for pointing this out and have revised the sentence as follows:

[Revised text] Since neutral excitons spread over the 2D crystal with fully in-plane electric dipole moment,

4) In Fig. 3b, there appears to be no quenching at $d = 0$ (tip in contact with the WSe₂). It is well known that radiation from emitters is usually quenched when in contact with metal surfaces (due to coupling to surface bound traveling plasmon-polaritons, electron-hole excitation in the metal (followed by non-radiative recombination, etc). What do the authors attribute this lack of quenching to?

Since we used the thin gold film (~10 nm thick) as a substrate for a TMD monolayer, we expect the far-field PL intensity ($d > 15$ nm) to be decreased due to nonradiative quenching. But, when the Au tip approaches the TMD monolayer with a few nm gap, we expect the suppressed

PL quenching because the spontaneous emission is coupled to the antenna mode with its fs-radiative decay [*Nano Lett.* **14**, 5270 (2014), *ACS Photon.* **5**, 186 (2018), *Nat. Nanotech.* **13**, 59 (2018)].

For the far-field PL quenching by the thin gold film, we recently performed control experiment with MoS₂ monolayers. When a MoS₂ monolayer is transferred onto the thin gold film, we expect that there is a thin water layer between them. As shown in the figure below, when the excitation laser power was 3 μW, PL intensity of the TMD crystal was not changed. On the other hand, when we used the high excitation power of >100 μW, the PL intensity was gradually decreased with respect to time up to ~50 %. From these results, we guess that the water layer was evaporated by the high-power excitation which leads to the increased PL quenching by the reduced distance between the gold film and the TMD crystal. To clarify the PL quenching property in our experiment, we have added this discussion in the Supplementary Information.

Fig. S11. (a, b) PL spectra of a MoS₂ monolayer with respect to the excitation time, exhibiting the PL quenching properties at the high excitation power. The CVD-grown MoS₂ monolayer is transferred onto the thin gold film through a wet-transfer method. We assume that there is a water layer between the TMD crystal and the gold film which can suppress the PL quenching phenomenon. When we use an excitation laser power of 3 μW, PL intensity is not changed (a). On the other hand, when use an excitation laser power of >100 μW, PL intensity is gradually decreased due to the PL quenching (b).

5) In the 2nd sentence on pg. 8, the authors state: "...as well as the largest plasmon-enhanced PL intensity of a transition metal dichalcogenide (TMD) monolayer reported to date [22]." This seems unprovable, and likely not true. For example, strain localized emitters with out-of-plane dipoles are likely enhanced by a larger factor, due to the mode polarization profile of tip-substrate gap mode. The authors should remove this claim from the text and the abstract.

We thank the reviewer for pointing this out. Based on the reviewer's comment, we have removed the sentence from the Abstract and the main text.

REVIEWERS' COMMENTS

Reviewer #1 (Remarks to the Author):

The authors satisfactorily responded to my comments.

Reviewer #2 (Remarks to the Author):

Since all remarks have been addressed this version can be recommended for publication in Nat. Comm.

Although some misprints and uncertainties should be considered, for example, '...warfront shaping' should change for wavefront shaping. Also, the spectral resolution depends not on a focused beam size, as the authors have claimed, but a diffraction grating only, whereas the size of the laser beam waist affects the spatial resolution indeed. Please, check your estimations of 39.8 cm⁻¹ and 7 cm⁻¹ for 150 g/mm and 1200 g/mm once again, it should be better in practice.

Reviewer #3 (Remarks to the Author):

I have read through the authors' responses and new files. I find their replies to be thoughtful, and that the changes generally address the comments and questions. I feel that the additions strengthen the paper; specifically, the added discussion of the applicability to differently fabricated tips and the additional data on quenching aspects of apertureless nano-imaging. I am excited to see others utilize their newly described approach in future near-field studies. Thus, the revised manuscript is recommended for publication in Nature Communications.

Point-by-point response:

Reviewer #1:

The authors satisfactorily responded to my comments.

We thank the reviewer for many constructive comments to improve our paper during the revisions.

Reviewer #2:

Since all remarks have been addressed this version can be recommended for publication in Nat. Comm.

Although some misprints and uncertainties should be considered, for example, ‘...warfront shaping’ should change for wavefront shaping. Also, the spectral resolution depends not on a focused beam size, as the authors have claimed, but a diffraction grating only, whereas the size of the laser beam waist affects the spatial resolution indeed. Please, check your estimations of 39.8 cm⁻¹ and 7 cm⁻¹ for 150 g/mm and 1200 g/mm once again, it should be better in practice.

We thank the reviewer for many constructive comments to improve our paper during the revisions. We have carefully checked our manuscript again to revised the typos. We also thank the comment on the spectral resolution. We have re-confirmed the estimated values in our setup several times and they were correct.

Reviewer #3:

I have read through the authors’ responses and new files. I find their replies to be thoughtful, and that the changes generally address the comments and questions. I feel that the additions strengthen the paper; specifically, the added discussion of the applicability to differently fabricated tips and the additional data on quenching aspects of apertureless nano-imaging. I am excited to see others utilize their newly described approach in future near-field studies. Thus, the revised manuscript is recommended for publication in Nature Communications.

We thank the reviewer for many constructive comments to improve our paper during the revisions.